# Involvement of miR-142 and miR-155 in Non-Infectious Complications of CVID

**DOI:** 10.3390/molecules25204760

**Published:** 2020-10-16

**Authors:** Giuliana Amato, Federica Vita, Paolina Quattrocchi, Paola Lucia Minciullo, Giovanni Pioggia, Sebastiano Gangemi

**Affiliations:** 1Operative Unit and School of Allergy and Clinical Immunology, Department of Clinical and Experimental Medicine, University of Messina, 98125 Messina, Italy; giulianaamato89@hotmail.it (G.A.); federica.vita75@gmail.com (F.V.); pquattrocchi@unime.it (P.Q.); pminciullo@unime.it (P.L.M.); gangemis@unime.it (S.G.); 2Institute for Biomedical Research and Innovation (IRIB), National Research Council of Italy (CNR), 98164 Messina, Italy

**Keywords:** common variable immunodeficiency, microRNAs, miR-142, miR-155, autoimmunity, cancer

## Abstract

Background and objectives: Common variable immunodeficiency (CVID) is the most prevalent antibody impairment. It is characterized by failure in immunoglobulin and protective antibody generation and defined by an increased tendency toward bacterial infections, autoimmunity, and malignancy. Most CVID diagnoses do not follow a classical Mendelian pattern of inheritance. In recent years, CVID has been considered an epigenetic phenomenon in the majority of cases, overtaking previous monogenetic and/or polygenetic theories. The aim of this study was to review the role of microRNAs (miRNAs) in CVID, focusing on the involvement of the same miRNAs in various non-infectious clinical complications of CVID, mainly autoimmunity and/or cancer. Materials and Methods: A bibliographic search of the scientific literature was carried out independently by two researchers in scientific databases and search engines. The MeSH terms “microRNAs” and “common variable immunodeficiency” were used. All research articles from inception to May 2020 were considered. Results: The literature data showed the involvement of two miRNAs in primary immunodeficiency: miR-142 and miR-155. Both of these miRNAs have been investigated through mice models, in which miR-142 and miR-155 were deleted. These knock-out (KO) mice models showed phenotypic analogies to CVID patients with hypogammaglobulinemia, adaptive immunodeficiency, polyclonal proliferation, lung disease, and enteric inflammation. miR-142 and miR-155 have been found to be involved in the following autoimmune and neoplastic clinical complications of CVID: Gastric cancer, gastric mucosa-associated lymphoid tissue (MALT) lymphoma, natural killer/Tcell lymphoma (NKTCL), and immune thrombocytopenia. Conclusions: miR-142 and miR-155 deregulation leads to similar CVID phenotypesin KO mice models. Although no data are available on the involvement of these miRNAs in human CVID, their dysregulation has been detected in human CVID comorbidities. The literature data show that miRNA sequences in murine models are comparable to those in humans; therefore, miR-142 and miR-155 involvement in human CVID could be hypothesized.

## 1. Introduction

Common variable immunodeficiency (CVID) is the most prevalent antibody impairment, characterized by failure in immunoglobulin and protective antibody generation, defined by an increased tendency toward bacterial infections, autoimmunity, and malignancy. Among the non-infectious complications (Figure 1), autoimmune diseases develop in 20–30% of subjects; immune thrombocytopenia purpura (ITP) and autoimmune hemolyticanemia are the most common, while other autoimmune features include inflammatory bowel disease, antiphospholipid syndrome, autoimmune thyroiditis, uveitis, primary biliary cirrhosis, vasculitis, and joint manifestations resembling rheumatoid arthritis. Regarding malignancy, gastric cancer and lymphoma are the most frequent findings; in particular, most lymphomas are Bcell-type non-Hodgkin lymphoma and some of these could be associated with mucosa-associated lymphoid tissue (MALT). Hodgkin lymphoma, chronic myeloid leukemia, thyroid papillary carcinoma, and pancreatic neuroendocrine carcinoma have also been described [1,2]. Most diagnoses do not follow a classical Mendelian pattern of inheritance. In recent years, CVID has been considered an epigenetic phenomenon in the majority of cases, overtaking previous monogenetic and/or polygenetic theories. Epigenetics results in changes in gene expression, without modifying the germ-line DNA gene sequences. Epigenetic mechanisms are influenced by various factors, such as infectious, environmental, nutritional, and iatrogenic stimuli, and include DNA methylation, cell-specific transcription factor expression, histone and chromatin modification, and non-coding RNAs. Early hematopoietic stemcell differentiation is susceptible to epigenetic processes, particularly DNA hypomethylation myeloid (as opposed to lymphoid) lineage engagement.

microRNAs (miRNAs) are small single-stranded non-coding RNA molecules that epigenetically regulate gene expression, mainly through binding to the 3′ untranslated region (3′-UTR) of targeted messenger RNA (mRNA) molecules and, thus, mediating translational repression, commonly along with mRNA degradation. The activity and function of key signaling pathways and cellular processes such as cell differentiation, proliferation, apoptosis, and response to hypoxia are controlled by a thousand different miRNAs, administering almost 60% of the protein-coding genes [3].

The aim of this study was to review the role of miRNAs in CVID, focusing on the involvement of the same miRNAs in various non-infectious clinical complications of CVID, mainly autoimmunity and/or cancer.

## 2. Materials and Methods

A bibliographic search of the scientific literature was carried out in scientific databases and search engines independently by two researchers. The MeSH terms “microRNAs” and “common variable immunodeficiency” were used. All research articles from inception to May 2020 were considered.

## 3. Results

The literature data show the involvement of two miRNAs in primary immunodeficiency: miR-142 [4] and miR-155 [5]. Both of these miRNAs have been investigated through mice models in which miR-142 and miR-155 were deleted. These knock-out (KO) mice models showed phenotypic analogies to CVID patients with hypogammaglobulinemia, adaptive immunodeficiency, polyclonal proliferation, lung disease, and enteric inflammation [6]. Although miR-142 and miR-155 deletion in CVID patients has not been detected yet, both of these miRNAs have been found to be involved in the following autoimmune and neoplastic clinical complications of CVID: Gastric cancer, gastric MALT lymphoma, natural killer/Tcell lymphoma (NKTCL), and immune thrombocytopenia (Figure 2). The potential pathways involving these two miRNAs in non-infectious complications of CVID are summarized in Figure 3.

### 3.1. miR-155 Expression in Pathophysiological Processes

miR-155-5p is expressed in many tissue and cell types and may play a fundamental role in various biological processes, including immune responses, inflammation, and hematopoiesis, and in different types of cancers, such as lymphomas, leukemias, and solid tumors. In contrast, low levels of miR-155-3p are present in hematopoietic cells. Recent studies have focused on the potential function of miR-155-3p underlying its cooperative role, mainly in the early stages of plasmacytoid dendritic cell activation, followed by miR-155-5p expression in the later stages [3]. Moreover, a tumor suppressive profile has been detected for miR-155-3p. The isoform –3p is silenced by hypermethylation of its host gene MIR155HG, thus promoting lymphomagenesis [7]. Moreover, MIR155HG expression is hardly linked to innate immunity and inflammation through the activation of both AP-1- and NF-kB-mediated mechanisms. Studies have shown that *Helicobacter Pylori* infection leads to miR-155-5p upregulation in human gastric epithelial cell lines and gastric mucosal tissues, as well as in primary macrophages derived from murine bone marrow. Co-incubation of human mesangial cells with INF-gamma and TNF cytokines also leads to an upregulation of miR-155-5p. This overexpression of miR-155-5p suppresses TAB2, which has beenpreviously detected as a direct target of this miRNA in myeloid cells and represents a fundamental signaling molecule in inflammatory reactions [3].

### 3.2. Potential Involvement of miR-142 in CVID Pathogenesis and Comorbidities

The micro-RNA 142 is evolutionarily conserved between human and murine species, allowing the transposition of studies on murine models to human clinical use [8]. In 2004, Chen et al. [9] showed that miR-142 is enriched in the following adult hematopoietic tissues: Bone marrow, the spleen, and the liver. At first, miR-142′s primary role was discovered to be essential in lineage differentiation in hematopoietic tissue, acting upstream of the hematopoietic transcription factorsFli, Gata2, Etvs, and Scl [10]. Over the years, its role has become known to be crucial for immune and inflammatory responses. Wang et al. [11], in 2012, demonstrated that an abnormal expression of miR142-3p leads to the blockage of myeloid differentiation, which are typical features of acute myeloid leukemia (AML). The role of miR-142 in immune homeostasis and its involvement in modulating the immunosuppressive response in tumor-associated macrophages was evaluated by Xu et al. in 2014. Macrophages are recruited to the neoplastic environment, where they assume a tumor-supportive phenotype, which is regulated by cytokines such as TGF-β. In this study, they showed a downregulation in the levels of miR-142 in immunosuppressive macrophages. The immunosuppressive macrophages were taken to apoptosis through the TGFBR1 inhibition caused by miR-142-3p overexpression [12]. Paradoxically, the overexpression of miR-142-5p targeting Smad3 has been shown to lead to cancer progression through the suppression of TGF-β-induced growth inhibition in cancer cells [10]. miR-142-5p is the predominant isoform expressed in thymically derived Tregs and determines peripheral immune tolerance suppressing self-reactive peripheral T effector cell (Teff) responses, thus limiting the development of autoimmunity activity and the upregulation of PDE3B, causing reduced intracellular cAMP and a failure of peripheral tolerance [13].

### 3.3. Dual Role of miR-155 and miR-142 in Gastric Cancer

Patients affected by gastric cancer (GC) that show low levels of miR-124-3p, miR-146a-5p, miR-335-5p, and miR-155-5p are characterized by major lymph node involvement, minor differentiation, and the presence of metastasis [14]. Zare et al., in 2019, elaborated on the expression of various miRNAs (e.g., miR-15b, miR-15a, and miR-186), including miR-155, in GC human tissue. They revealed low levels of miR-155-5p in GC. Even though there is conflicting evidence about the role of this miRNA in human cancer, recent studies strongly highlight its tumor suppressor role. Supporting these findings, the deregulation of miR-155-5p starts in the early stages of gastric carcinogenesis, similar to *H. pylori*-induced gastritis and gastric dysplasia. Moreover, the expression of miR-155-5p is heavily repressed in tumors at advanced stages and in lymph node metastasis [15]. A direct target of miR-155 is cyclin D1, which can act as an oncogene in GC. Previously, high levels of this cell cycle regulator have been identified in various human malignancies, including GC [16]. Inhibition of miR-155-5p leads to the transformation of bone marrow mesenchymal stem cells (MSCs) into GC humantissue-derived MSC-like cells through NF-kB p65 activation; on the contrary, overexpression represses GC cell proliferation, thus stimulating their apoptosis. miR-155-5p targets MAP3K10 and consequently inhibits the proliferation of neoplastic cells [17]. miR-155 can neither be evaluated as an oncomiR nor as a neoplastic-suppressor miRNA, suggesting that it may regulate its target transcript in a dose-dependent manner. An antitumor response of adaptative immunity is the production of INF-gamma. miR-155 targeting SHIP1 leads to the development of IFN-γ-producing CD4^+^ and CD8^+^ lymphocytes. Moreover, miR-155 in CD8 T cells decreases suppression of cytokine signaling 1 (SOCS1) activity, leading to an improved anticancer response, thus using both innate and adaptative immune cells [18]. Ji et al. showed that miR-155 improves the antitumor response by epigenetically limiting CD8^+^ T cell differentiation and activity depletion [19].

Primarily, miR-142-5p acts as an oncomiR through the TGFβ signaling pathway, targeting SMAD3. In contrast, in GC, its suppression leads to the major proliferation of neoplastic cells. miR-142-5p targets CYR61, the pivotal element of the nuclear translocation of β catenin, thus inhibiting the WNT pathway [20]. miR-142-3p has been described to be downregulated in 100 pairs of GC tissue compared to non-tumor tissue. Wang et al. identified cyclin T2 as a direct target of this miRNA to control the migration, invasion, and proliferation of gastric cancer cells. CCNT2 has a vital role in transcription initiation and elongation driven by RNA polymerase II. Although the role of CCNT2 in carcinogenesis is poorly understood, it has been reported that this target could enhance the proliferation of ThT1 cells and could repress monocytic differentiation [11,21].

### 3.4. Immunotolerance and Antitumor Immunity: The Involvement of the PD-L1/PD-1 Axis in Malignancies

miR-155 has a key role in preventing excessive immunosuppression through the control of PD-L1 levels, following TNF and IFN-γ stimulation [18]. PD-L1 is also the potential target gene of miR-142-5p. Jia et al. [22], firstly, in 2017, detected this balance from samples of human large B cell lymphoma, thymoma, and pancreatic cancer. In tissue cells, PD-L1 mediates peripheral T cell tolerance; in cancer cells, it promotes immune evasion linking PD-1 on T cells. Low levels of miR-142-5p are related to poor survival in gastric cancer patients; consequently, high levels of PD-L1 in cancer cells are associated with poor prognosis. miR-142-5p regulates PD-L1 expression through binding to its 3′ untranslated region (UTR). The overexpression of miR-142-5p leads to decreased mRNA and protein levels of PD-L1. Therefore, miR-142-5p expression, as shown by Jia et al., is inversely correlated with PD-L1. Thus, a blockade of the PD-L1/PD-1 pathway enhances antitumor immunity by upregulating tumors that are infiltrating T lymphocytes and are decreasing PD-1^+^ T lymphocytes. TNF-α, IFN-γ, and IL-10 are three cytokines related to the PD-L1/PD-1 pathway. miR-142-5p overexpression leads to the upregulation of TNF-α and IFN-γ, which have antitumor effects, and the downregulation of IL-10, which has immunosuppressive functions [22].

### 3.5. Over the Chronic Gastritis to Overt Gastric MALT Lymphoma

Blosse et al. found a different pattern of dysregulated miRNAs (the upregulation of miR150, 155, 138, and 196-a and the downregulation of miR-7 and miR-153) in human GML samples. The upregulation of miR-155 during Hp infection in MALT lymphoma is related to a worse response to eradication therapy. NF-kB and MYB directly activate the expression of miR-155 in B cells, provoking the unblocking of AKT activity and influencing BCR signaling. NF-kB and BCR activation plays a key role in MALT lymphomagenesis [23]. Former studies have shown the upregulation of miR-142 and miR-155 when comparing human GML tissue to normal mucosa [24] and chronic gastritis [25]. Both miR-142 and miR-155 inhibit the pro-apoptoticstress-induced P53 target gene (TP53INP1), thus accelerating MALT lymphoma proliferation [23]. Fernandez et al. proposed the use of the quantitative expression of miR-142-3p and miR-155 to better distinguish between chronic gastritis and gastric MALT lymphoma. Finally, in the disseminate presentation of gastric MALT lymphoma, among 23 miRNAs, miR-142-3p and miR-155 were found to be downregulated [25]. Several studies about the prognostic value of miR-155 expression have been conducted. Bedewy et al., despite contradicting results in the literature, affirmed that only a low IPI score (prognostic index) and low miR-155 expression are predictors of longer event-free survival in B cell NHL patients [26].

### 3.6. Lymphangiogenesis Arrangement: The Role of miR-155

Cuadros et al., in 2018, detected BRG1 and miR-155 expression in 23 human leukemia cell lines. BRG1 is a fundamental regulator of leukemia stem cells. Commonly, BRG1 has a tumor suppressor role in solid tumors, even if some studies have reported high levels in prostate, skin, and gastric cancer, which is related to neoplastic progression. Moreover, several findings have shown that BRG1 is needed for leukemia cell propagation, preserving a phenotype that controls MYC proto-oncogene expression. In their previous study, it was found that a lack of BRG1 in lung tumors could be related to the action of different miRNAs, such as miR-155. The authors, in 2018, found that BRG1 levels were inversely proportional to miR-155 expression in Burkitt’s lymphoma and the diffuse large B cell lymphoma human cell line [27]. Chang et al., in 2019, reported that miR-155 regulates lymphangiogenesis in NKTCL through the miR-155/BRG1/STAT3/VEGFC pathways in human NKTCL cell lines. They demonstrated, using antagomiR-155, pro-lymphangiogenesis capability limitations and the induction of apoptosis of NKTCL cells in vitro, as well as neoplastic growth reduction in vivo. By targeting miR-155, BRG1 levels can be restored, and VEGFC is consequently repressed. On the other hand, they showed that this axis does not control VEGFD, the levels of which were not found to be repressed, underlining the pivotal role of lymphangiogenesis in malignancies of NKTCL [28].

### 3.7. Dysregulation of Peripheric Immunotolerance and Immune Thrombocytopenia

miR-142-5p plays a key role as an immunometabolic regulator of intracellular cAMP, targeting phospodiesterase-3b (Pde3b) mRNA and controlling the suppressive function of Tregs. Pde3b gene hydrolyzes its substrate cAMP to AMP. Teffs need low cAMP levels for their activation, so Tregs try to maintain high levels of intracellular cAMP. FOXP3 and miR-142-5p work together to maintain pde3b suppression; in the absence of miR-142-5p, FOXP3-related repression of pde3b is not adequate to prevent the remarkable activity and upregulation of the Pde3b gene, causing reduced intracellular cAMP and a failure of peripheral tolerance. Treg lineage is regulated by other miRNAs, which target and inhibit key components of the PI3K/AKT/MTOR signaling network, favoring Treg induction over Teff production. For example, miR-155 upregulates the production of Tregs through the SOCS1. First, Anandagoda reported that miR-142 plays a cell-intrinsic positive role in reinforcing Treg repressive activity [13]. Previously, Huang et al. demonstrated that Teff maintains high levels of the other isoform –3p to limit adenylyl cyclase 9 (AC9) expression, an enzyme fundamental for cAMP production in CD4^+^Tcells, thus restricting endogenous cAMP generation. These findings confirm that the differential expression of miR-142 isoforms preserves TregcAMP intracellular concentration. The generation of cAMP (low miR-142-3p, AC9 augmented), as well as the suppression of its hydrolysis to AMP (high miR-142-5p, Pde3b gene repressed), could therefore express the parallel factors of the same molecular mark [29].

Although the immune thrombocytopenia (ITP) molecular process remains unknown, an imbalance of the Th subset (Th1, Th17, and Th22) could be associated with the dysregulation of the miRNA profile. Previous studies have shown high levels of Th1, Th22, and Th17 cells, while Treg (CD4^+^, CD25^+^, and Foxp3^+^) cells are suppressed in the peripheral blood of ITP patients [30]. Autoimmunity and severe inflammation may be observed in humans and mice with aberrant Tregs, highlighting that Tregs could help to limit autoimmune responses. The association between miRNAs and ITP is still unclear, but miRNAs maintain an immune function and inhibit the inflammatory reaction in Tregs. Zhu et al. identified 11 miRNAs downregulated and 26 miRNAs upregulated in the Tregs from ITP patients compared to healthy controls. Among the downregulated miRNAs, they focused their attention on miR-155-5p, miR-146b-5p, and miR-142-3p.miR-155downregulation stimulates CD4^+^Th cells for Treg-mediated suppression. Indeed, former studies have reported that miR-155 knockout mice present low levels of Tregs, both in the periphery and in the thymus. miR-142-3p also influences Treg function through the regulation of cAMP production targeting AC9 in Tregs. miR-142-3p prevents cAMP levels in Tregs from suppressing AC9 generation, while Foxp3 inhibits miR-142-3p from maintaining the activation of the AC9/cAMP pathways in Tregs [31].

## 4. Discussion

The results of this study underline the involvement ofmiR-155 and miR-142in biological processes such as hematopoiesis, inflammation, immune responses, and carcinogenesis, therefore indicating that their absence or deregulation could leadto these pathologies, accordant with most comorbidities found in CVID patients.

In CVID, approximately half of the patients present with non-infectious complications, including autoimmune, lung, and gastrointestinal diseases, benign lymphoproliferation, and malignancies. CVID patients have an approximately 47-fold increased risk of developing gastric cancer and a 30-fold increased risk of lymphoma [32].

Nowadays, molecular changes in precancerous and tumor tissues have assumed a pivotal role in studies on carcinogenesis. Low levels of miR-155-5p and -142 have been revealed in GC [15,21,22]. Supporting these findings, the deregulation of miR-155-5p starts in the early stages of gastric carcinogenesis, such as *H. pylori*-induced gastritis and gastric dysplasia. Moreover, the expression of miR-155-5p is heavily repressed in tumors in advanced stages and in lymph node metastasis [15]. miR-155 regulates the production of INF-gamma, leading to a better anticancer response and preventing excessive immunosuppression [18]. Because most GC cases are diagnosed in advanced stages, Ma et al. proposed a potential diagnostic biomarker function of miR-155 to detect the early stages of GC [16].

Several studies have reported an increased risk of non-Hodgkin lymphoma in CVID patients. MALT lymphoma, a subtype of low-grade B cell NHL, seems to be a result of autoimmunity or chronic inflammatory stimulation in lymphoid tissue. In the stomach, the colonization and consequent infection by *H. pylori* can induce mucosal inflammation (gastritis) and, later, drive advancement to gastric MALT lymphoma. In recent years, the role of microRNAs has been investigated in gastric MALT lymphomagenesis. The miRNA expression profile has been compared with human samples of MALT lymphoma, chronic gastritis, and normal lymphoid tissue. miR-142-3p and miR-155 were found upregulated in gastric MALT lymphoma according to previous data, in which these miRNAs played a crucial role in MALT lymphomagenesis. Additionally, lymphangiogenesisis regulated by miR-155 in the human NKTCL cell line [28]. Moreover, a worst response to HP eradication therapy in MALT lymphoma is related to the upregulation of miR-155 [23]. In contrast, in metastatic GML, both of these miRNAs have been found to be downregulated [25]. Bedewy et al. validated the previous data, confirming the prognostic value of low miR-155 expression in B cell NHL patients [26].

A significant number of CVID patients present with autoimmunity issues. According to our results, miR-142 and miR-155 are critical for the organization of Treg-mediated peripheral tolerance and the prevention of autoimmunity through the direct repression of Teffs. T helper cells are crucial in the development and advancement of inflammatory and autoimmune diseases. Particularly, autoimmune thrombocytopenia is the most frequent, occurring in approximately 16% of CVID patients. The dysregulation of the miRNA profile and, consequently, the imbalance of the Th subset could be implicated in the pathogenesis of ITP [30]. Among non-infectious complications, CVID patients show an increased risk of splenomegaly. miR-142 plays a critical role in controlling the network of erythrocyte proteins required to maintain a typical cellular morphology and function, and for oxidativestress defense. The miR-142KO mouse line shows predominant extramedullary erythropoiesis, exhibiting splenomegaly and disorganized splenic parenchyma, partly due to the excessive destruction of red cells [33]. Moreover, the loss of miR-142 causes an extension of the spleen and an immune proliferativedisorder, preferring Bcell and myeloid cell populations to the detriment of functionality, caused by defects in B cell maturation and differentiation. miR-142 KO mice present an enlarged splenic Bcell compartment, above all due to the expansion of MZ-like B cells and the contraction of T and B1 Bcell populations in the periphery [4].

## 5. Conclusions

In conclusion, miR-142 and miR-155 deregulation is present in murine models of CVID. No data are present on the role of these two miRNAs in human CVID. However, such a deregulation, investigated through human cell lines, has been also detected in various pathologies, usually considered as non-infectious CVID complications.

The literature data show that miRNA sequences in murine models are comparable to those in humans [8,34]. This led some authors to hypothesizing a translation of the results found in mice to humans. Moreover, studies on autoimmune diseases have reported the involvement of the same miRNAs in murine models and humans [35,36]. In light of these findings, the involvement of miR-142 and miR-155 in human CVID could also be hypothesized. However, this thesis needs to be proven through exvivo studies on subjects affected by CVID.

Moreover, it would be interesting to explore miR-155 function with respect to disease outcome. This miR, indeed, shares the same role as IL-21 in B cell differentiation and could lead to two different phenotypes: One with typical recurrent infections and a better outcome, and one characterized by non-infectious complications, such as splenomegaly and autoimmune and neoplastic diseases, as found in IL-21 KO mouse models [37,38].

## Figures and Tables

**Figure 1 molecules-25-04760-f001:**
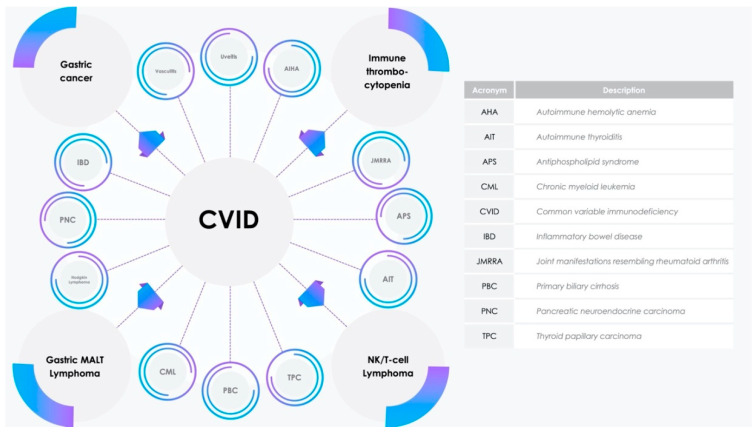
Non-infectious complications of common variable immunodeficiency (CVID). MALT, mucosa-associated lymphoid tissue; NK, natural killer.

**Figure 2 molecules-25-04760-f002:**
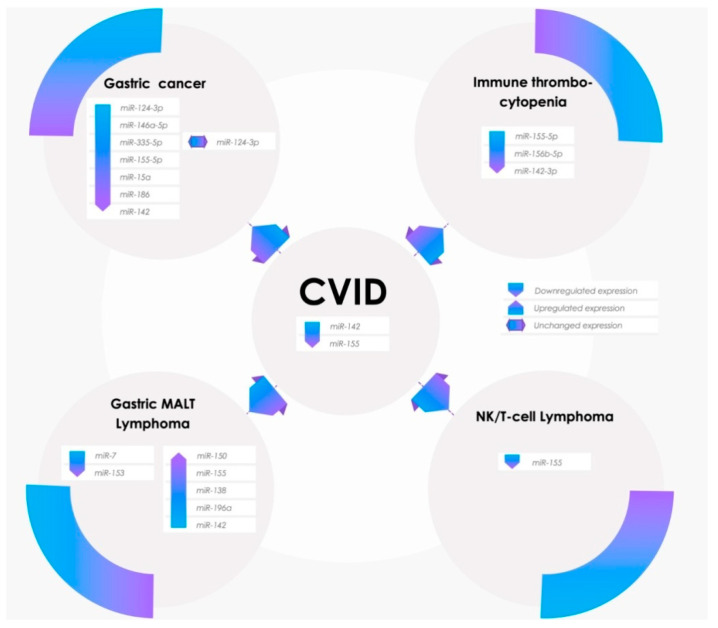
Non-infectious complications of CVID and microRNAs (miRNAs).

**Figure 3 molecules-25-04760-f003:**
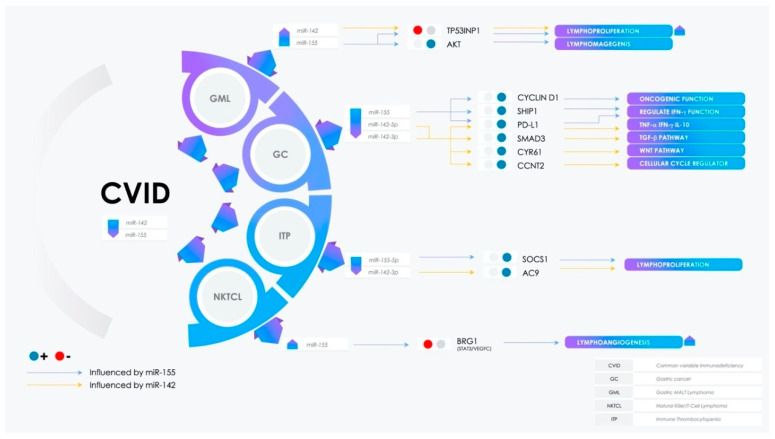
Potential pathways involving miR-142 and miR-155 in non-infectious complications of CVID.

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
