# Peer review of "Involvement of miR-142 and miR-155 in Non-Infectious Complications of CVID"

_molecules, 2020, doi:10.3390/molecules25204760_

Round 1

Reviewer 1 Report

This is an interesting mini-review to summarize the potential role of microRNAs in CVID. Through literature search and publication comparison, authors focused on two miRs in this context, miR-142 and miR155.  Phenotypes of mouse models with miR142 or miR155 ablation were further described to establish their potential link to CVID. Although authors did not provide a comparison between mouse model and human observations, such as the pros and cons of observations in these mouse models relative to CVID, unilaterally forcing the similarity comparison of results from mice or cell lines to human CVID disease may be misleading for future studies. It would be helpful to also discuss the challenge for such  comparison that may need further investigation.  

Author Response

We thank the reviewer for the comments and suggestions. We improved the text accordingly. 

Reviewer 2 Report

Common variable immunodeficiency (CVID) is the greatest prevalent antibody impairment characterized by failure in immunoglobulin and protective antibody generation, defined by an increased tendency toward bacterial infections, autoimmunity, and malignancy. In this manuscript, the authors provide a detailed discussion on the roles of miRNAs in CVID, focusing on the involvement of miR-142 and miR-155 in various non-infectious clinical complications of CVID, mainly autoimmunity and/or cancer. The manuscript is well written and the illustration is presented in a good quality. This will provide interesting information for the reader of the journal. However, there are still many grammatical and syntax errors in the article. So I think the manuscript can be accepted after grammar and language check.

Author Response

We thank the reviewer for the comments. We performed grammar and language check. 

This manuscript is a resubmission of an earlier submission. The following is a list of the peer review reports and author responses from that submission.